# Changes in the Quality Attributes of Selected Long-Life Food at Four Different Temperatures over Prolonged Storage

**DOI:** 10.3390/foods11142004

**Published:** 2022-07-06

**Authors:** Tomáš Šopík, Zuzana Lazárková, Richardos Nikolaos Salek, Jaroslav Talár, Khatantuul Purevdorj, Leona Buňková, Pavel Foltin, Petra Jančová, Martin Novotný, Robert Gál, František Buňka

**Affiliations:** 1Department of Food Technology, Faculty of Technology, Tomas Bata University in Zlín, Nam. T.G. Masaryka 5555, 760 01 Zlin, Czech Republic; sopik@utb.cz (T.Š.); rsalek@utb.cz (R.N.S.); gal@utb.cz (R.G.); frantisek.bunka@gmail.com (F.B.); 2Centre of Polymer Systems, University Institute, Tomas Bata University in Zlín, Tr. T. Bati 5678, 760 01 Zlin, Czech Republic; 3Laboratory of Food Quality and Safety Research, Department of Logistics, Faculty of Military Leadership, University of Defence, Kounicova 65, 662 10 Brno, Czech Republic; jaroslav.talar@unob.cz (J.T.); pavel.foltin@unob.cz (P.F.); martin.novotny5@unob.cz (M.N.); 4Department of Environmental Protection Engineering, Faculty of Technology, Tomas Bata University in Zlín, Nam. T.G. Masaryka 5555, 760 01 Zlin, Czech Republic; purevdorj@utb.cz (K.P.); bunkova@utb.cz (L.B.); jancova@utb.cz (P.J.)

**Keywords:** long-term storage, food quality, food safety, long-life food, temperature regimens

## Abstract

This study reports the development of selected indicators affecting changes in food quality and safety of selected long-life canned (Szeged goulash, canned chicken meat, pork pâté, canned tuna fish) and dehydrated (instant goulash soup) food during a two-year storage experiment at four different temperatures. The storage temperatures were selected to represent Arctic (−18 °C), temperate (5 °C), subtropical (25 °C) and tropical (40 °C) climatic zones where such food is likely to be stored during, for example, humanitarian and military missions. Microorganism amounts below the detection limit (*p* < 0.05), regardless of the storage temperature (*p* ≥ 0.05), were monitored in canned samples. The contents of dry matter, fat and proteins did not change during storage, regardless of the storage temperature (*p* ≥ 0.05). During the 24-month storage, all food showed an increase in the level of ammonia (*p* < 0.05) and the TBARS-value (*p* < 0.05), whereas the rate of increase in both parameters was significantly higher at higher storage temperatures (*p* < 0.05). The losses of individual amino acids during storage ranged from 5% rel. calculated on the amino acid contents in Month “0” up to 15% rel. (*p* < 0.05). With storage temperatures above the freezing point, the hardness values decreased with the increase in the storage temperature (*p* < 0.05) and prolongation of the storage period (*p* < 0.05). Moreover, with temperatures of −18 °C, the development of hardness, measured as the “decrease rate”, was significantly higher compared to the absolute values.

## 1. Introduction

The North Atlantic Treaty Organization (NATO) is currently engaged in many foreign humanitarian or military missions all over the world. Increased attention must be paid to the diet of such mission members since possible malnutrition could result in a decrease in their action or combat readiness; moreover, the missions usually take place in subtropical and tropical climates where food safety is of great importance [1]. Most food used in missions is canned or dehydrated due to long transport distances, again mainly in subtropical and tropical climates. Such food also includes cans containing various types of meat or fish products. Therefore, it is necessary to use preservation methods that ensure the inactivation not only of vegetative forms of microorganisms but also of bacterial spores. These preservation methods include the thermal sterilisation of packed food (aluminium or iron-based packaging). Dehydration processes can also be utilised when the water activity is reduced to values that do not allow the development of microflora (especially dehydrated soups). In addition to providing for the diet of foreign mission members, these components can be used for innovative combat food portions (a food package for 24 h). Here, the storage life condition is at least 24 months at 25 °C. Subtropical and tropical climates have been primarily mentioned. However, NATO forces could also be deployed in Arctic zones [2,3,4,5,6,7].

During storage at temperatures of >25 °C and at <−18 °C, a variety of mutual intensive interactions of individual components may occur; these can worsen the food quality and, in some cases, even the safety of the food. In the case of proteins, amino acid chain depolymerisation can occur in a limited way, but Maillard reactions (reactions of carbonyl compounds with amino compounds, especially reducing carbohydrates and amino acids) are primarily in progress. The Strecker degradation may represent another reaction when ammonia is released. During these interactions, aromatic active substances are usually formed. Lipids are also subject to interactions during storage, primarily the sequence of autooxidation reactions that lead to the rancidity of the lipid fraction present. Carbohydrates especially enter Maillard reactions as an important reactant [8,9,10,11,12,13,14,15].

The above-mentioned examples of possible interactions also affect the organoleptic quality, mainly the contents of aromatic active substances. The textural properties of products can also be significantly affected [2,11,16,17,18]. Finally, the microbial profile of a given food must be dealt with. This can be substantially affected by the two-year storage even when using packaging with good barrier properties [19,20].

Currently, producers of canned and dehydrated products usually have products meant for temperature ranges from 0 to 25 °C tested within the hazard analysis (HACCP system). For these temperatures, the shelf-life of their products is stated on the packaging. Unfortunately, even the producers cannot provide information related to how the food might change if it is stored on a long-term basis at temperatures of >25 °C [2,3].

Therefore, the objective of this work for selected long-life canned and dehydrated food was to conduct a two-year-long storage experiment at temperatures of −18 °C, 5 °C, 25 °C and 40 °C representing the temperatures of Arctic, temperate, subtropical and tropical climatic zones. Foods that are actually used for the purposes described above were selected. Another goal was to monitor selected indicators affecting changes in food quality and safety, especially with a focus on changes in proteins, lipids and carbohydrate components, and then on organoleptic and microbiological properties of the monitored food. The selection of food included cans with pork and chicken meat and a fish product for comparison. As dehydrated food for comparison with cans, a goulash instant soup was selected. Therefore, the term “long-life food”, which, in a wider context, comprises all food used, is used throughout the work.

## 2. Materials and Methods

### 2.1. Samples and Samplings

For this study, the following long-life foods were chosen: (i) instant goulash soup (IGS; Hügli Food Ltd., Czech Republic, Zásmuky u Kolína, nutrition information from the packaging (per 100 g): fat 12.0 g, carbohydrate 52.0 g, of which sugars 18.0 g, protein 13.0 g); (ii) Szeged goulash (SG; Hamé Ltd., Czech Republic, Kunovice, nutrition information from the packaging (per 100 g): fat 9.5 g, carbohydrate 5.9 g, of which sugars 1.5 g, protein 4.3 g); (iii) canned chicken meat (CCM; Hamé Ltd., Czech Republic, Kunovice, nutrition information from the packaging (per 100 g): fat 5.6 g, carbohydrate 0.7 g, of which sugars 0.3 g, protein 17.2 g); (iv) pork pâté (PP; Hamé Ltd., Czech Republic, Kunovice, nutrition information from the packaging (per 100 g): fat 29.3 g, carbohydrate 1.5 g, of which sugars 0.6 g, protein 9.1 g) and (v) canned tuna fish (CTF; Gaston Ltd., Czech Republic, Kunovice, nutrition information from the packaging (per 100 g): fat 7 g, carbohydrate 0.5 g, of which sugars 0.5 g, protein 18.0 g). Samples were purchased as single-serving packages: IGS (109 × 20 g; sealed in cardboard/polypropylene/aluminium foil), SG (61 × 400 g; aluminium can), CCM (61 × 180 g; iron can), PP (61 × 100 g; aluminium can), CTF (61 × 80 g; iron can). All food products were purchased within one month from the date of manufacture. The manufacturers of the tested samples declare the minimum shelf life of most of these foods is two years (one year in the case of the IGS) and recommend storage at ambient temperature.

The storage experiment was conceived to represent four distinct climatic zones. Therefore, the examined samples were stored in a freezer (−18 ± 2 °C; simulating storage conditions in the Arctic zone), a refrigerator (5 ± 2 °C; used as a reference temperature), a controlled temperature chamber (25 ± 2 °C; simulating storage conditions in the temperate zone) and another controlled temperature chamber (40 ± 2 °C; simulating elevated temperature storage conditions in the sub/tropical zone). All foods were subjected to microbiological (see Section 2.2), chemical (see Section 2.3), textural (see Section 2.4) and sensory analyses (see Section 2.5), as well as determination of amino acids (see Section 2.6).

Generally, samples were taken immediately after purchase and then after 3, 6, 9, 12, 15, 18, 21 and 24 months of storage. All analyses were performed at the beginning of the storage experiment, marked as Month “0”, and the full above-mentioned sampling intervals for 5 °C, 25 °C and 40 °C were chosen. However, the sampling interval for −18 °C was longer, approximately a year, because we expected that changes would not be intensive. Deviations from this general scheme are described for each method. Three independent lots were sampled; during each test, three packages were sampled; each sample was analysed three times (3 lots × 3 samples × 3 analysis; *n* = 27). The exception to this was the sensory analysis, where the number of analyses was determined by the number of assessors (*n* = 24).

A sample was excluded from testing if it showed evidence of too-high microbiological activity (which could be harmful to human health) or if it failed the sensory assessment (unacceptable or excessive score, see Section 2.5).

### 2.2. Microbiological Analysis

The microbial quality of the stored foods was analysed by the assessment of the total number of microorganisms (CFU) according to ISO Standard No. 4833:2013 [21] and the number of aerobic and anaerobic spore-forming microorganisms according to Harrigan [22]. The colony-forming units of yeasts and/or moulds were assessed according to ISO Standard No. 21257-1:2008 [23] for SF, CCM, PP and CTF or according to ISO Standard No. 21257-2:2008 [24] for IGS. For 40 °C, an analysis after one month was included. Since stable canned samples were selected, the sampling plan was determined for the 3rd, 12th and 24th months of storage. All analyses were, at a minimum, performed in triplicate (*n* = 27).

### 2.3. Chemical Analysis

Dry matter content was assessed according to ISO 1442:1997 [25]. Fat and crude protein analysis was carried out according to the Soxhlet and Kjeldahl methods, according to ISO 1871:2009 [26] and ISO 17189:2003 [27], respectively. The samples for fat and crude protein were taken after 12 and 24 months since it was not assumed that the lipid fraction and the nitrogen contents (nitrogenous substances) would change. On the other hand, we expected that the representation of individual lipid fractions (e.g., lipid auto-oxidation products) would change, as well as compounds where nitrogen is bound (e.g., intermediate products and products of Maillard reactions and the Strecker degradations).

The ammonia content was assessed using the microdiffusive Conway method, as described by Buňka, Hrabě and Kráčmar [28]. Lipid oxidation was measured by the thiobarbituric acid reactive substances (TBARS) contents method previously described by Kristensen and Skibsted [11], and the absorbance units at the used wavelength per mg of sample (A_538_/mg) were used for result expression. All basic chemical analyses were performed at a minimum in triplicate (*n* = 27).

### 2.4. Textural Analysis

Textural analysis was used only in CCM and PP, as their texture allowed this type of analysis. Samples were measured in the original packaging by penetration test utilising the TA.XTplus texture analyser (Stable Micro Systems Ltd., Godalming, UK) with a plate aluminium probe (20 mm diameter). A software Exponent Lite (version 4.0.13.0; Stable Micro Systems Ltd., Godalming, UK) was used for data collection and expression as force–time curves. The following parameters were set up: 50% strain, trigger force 5 g, deformation rate 2 mm/s and temperature 25 ± 1 °C. The hardness values were calculated as the maximum force (N) observed during the penetration.

### 2.5. Sensory Analysis

Sensory evaluation of the stored samples was assessed at each sampling time by a panel consisting of 24 selected assessors (*n* = 24) trained according to ISO Standard No. 8586:2012 [29]. The IGS sample was prepared with water according to the producer’s instructions (given on the packaging). The samples were served in a sensory laboratory furnished with sensory booths in accordance with ISO 8589:2007 [30]. A controlled temperature of 20 ± 2 °C and normal light conditions were exploited, and samples were submitted in a random order to guarantee sample anonymity. In order to avoid carry-over effects, water for mouth rinsing between the samples was allowed. A seven-point scale (1—excellent, 4—good, 7—unacceptable) was used to determine the appearance, consistency and flavour, and a seven-point scale (1—negligible, 4—medium, 7—excessive) was utilised for off-flavour intensity. Each point of the scale was described with the use of terms for intensity scales.

### 2.6. Determination of Amino Acids

Fifteen amino acids (aspartic acid, threonine, serine, glutamic acid, proline, glycine, alanine, valine, isoleucine, leucine, phenylalanine, tyrosine, histidine, lysine, arginine, methionine and cysteine) were analysed by ion-exchange chromatography as described by Buňka et al. [31]. During the acid hydrolysis, asparagine and glutamine were transformed into aspartic and glutamic acid, respectively. Amino Acid Analyser AAA400 (Ingos, Prague, Czech Republic) with 370 mm × 3.7 mm column filled with an ion exchanger was used for the determination of amino acids. Furthermore, post-column ninhydrine derivatisation and spectrophotometric detection were employed. Sodium citrate buffers were used as a mobile phase. All amino acid standards (in HPLC grade purity) were obtained from Ingos, Prague, Czech Republic. Samples for amino acid determination were taken at the beginning of the experiment and then after 12 and 24 months. Each data set was used to calculate an essential amino acid index (EAAI; %). All analyses were, at a minimum, performed in triplicate (*n* = 27).

The Essential Amino Acid Index (EAAI; %) was calculated according to Equation (1).
(1)$$EAAI=\sqrt[{n}]{\frac{100\cdot A_{1}}{B_{1}}\cdot \ldots \;\cdot \frac{100\cdot A_{n}\;}{B_{n}}},$$

*A* is the amount of a specific amino acid in the test protein in g/100 g of crude protein.

*B* is the amount of the same amino acid in the reference protein in g/100 g of crude protein (whole milk protein).

### 2.7. Statistical Analysis

The results were evaluated using Kruskal–Wallis and Wilcoxon tests at 5% level of significance. Unistat^®^ 6.5 (Unistat Ltd., London, UK) was used for the statistical evaluation.

## 3. Results

### 3.1. Results of the Microbiological, Basic Chemical and Sensory Analyses

The results of the IGS microbiological analysis are stated in Table 1. During the two-year-long storage, the amounts of the total bacterial count, aerobic and anaerobic spore-forming bacteria (*p* < 0.05), rose. The growth was more intense at higher storage temperatures (*p* < 0.05). After six months of storage at 40 °C, the counts of these indicator groups of microorganisms were so high (*p* < 0.05) that IGS could not be assumed as harmless to health; therefore, IGS monitoring was terminated after six months. After storing IGS at 25 °C for 18 months, most analyses were again terminated due to high amounts of the total bacterial count; aerobic and anaerobic spore-forming bacteria (*p* < 0.05). The numbers of moulds and yeasts that fluctuated below the detection limit (*p* < 0.05) throughout the whole storage period were also analysed.

For other monitored foods (SG, CCM, PP and CTF), the total bacterial count, aerobic and anaerobic spore-forming bacteria, as well as counts of mould and yeasts throughout the whole storage period, were below the detection limit (*p* < 0.05), regardless of the storage temperature (*p* ≥ 0.05).

Table 2 shows dry matter, crude protein and fat content results of the five analysed foods during the two-year-long storage at four different temperatures. The results clearly show that the contents of these parameters have not changed during storage, regardless of the storage temperature (*p* ≥ 0.05). Moreover, as can be observed, the crude protein and fat content results do not significantly differ from the data stated by the producer on the packaging (*p* ≥ 0.05).

Appendix A in the Appendix A show the results of the organoleptic assessment of appearance, consistency, flavour and off-flavour using a seven-point intensity scale. In the case of IGS (Appendix A in the Appendix A), substantial deterioration of consistency, flavour and off-flavour appeared in the 6th and 9th months of assessment (due to the results of the microbiological analysis, the sample was not swallowed but removed from the oral cavity using sufficient rinsing); therefore, IGS stored at 40 °C was not further evaluated using sensory tests. Similarly, IGS stored at 25 °C was not assessed from the 21st month of storage. Defects in flavour and off-flavour also appeared at the storage temperature of −18 °C, when samples with a high degree of foreign odours and flavours were not acceptable after two years.

In the first year of storage below 40 °C, SG, PP and CTF products (Appendix A in the Appendix A) were usually assessed as excellent to good in the monitored organoleptic parameters. At the storage temperature of 40 °C, the organoleptic parameters were assessed as less good or even worse (*p* < 0.05). The deterioration appeared after the 12th to the 18th month of storage (*p* < 0.05) with the fact that the rate of deterioration of monitored properties at SG, PP and CTF significantly increased together with the increasing storage temperature (*p* < 0.05). 

The development of organoleptic properties of CCM (Appendix A in the Appendix A) differed from the other foods in that these products were microbiologically stable regardless of the storage temperature throughout the whole two-year storage (See Table 1). At the temperatures of −18 °C and 5 °C, the products were assessed as excellent to good in the first year of storage. Afterward, the organoleptic quality of products stored at these temperatures began to deteriorate (*p* < 0.05). A faster deterioration in flavour and off-flavour was noted at the storage temperatures of 25 °C and 40 °C (*p* < 0.05). When storing at 40 °C, the products were already unsatisfactory in the 9th month of storage, and at 25 °C, CCM became unsatisfactory in the 21st month.

Generally, in the case of storage at 25 °C and 40 °C, IGS and CCM monitoring was terminated in the above-mentioned months for microbiological or sensory reasons. However, especially for some chemical analyses (dry matter, crude protein, fat and amino-acid contents), the storage of these foods continued in order to complete the time series.

### 3.2. Results of Chemical Analysis Describing Protein and Lipid Degradation

Table 3 presents the development of the content of ammonia as an intermediate product or the final product of the reaction of nitrogenous substances and the TBARS-value showing the contents of lipid oxidation secondary products. The development of both parameters was monitored during the two-year storage at four different temperatures. During the 24-month storage, all five foods (IGS, SG, CCM, PP and CTF) showed an increase in the content of ammonia (*p* < 0.05) and the TBARS-value (*p* < 0.05), whereas the “growth rate” of the content of ammonia and the TBARS-value was significantly higher at higher storage temperatures (*p* < 0.05). Differences in the “growth rate” of the content of ammonia and the TBARS-value in time were also observed among certain foods. At temperatures below 40 °C, only gradual increases in the content of ammonia and the TBARS-value were observed for SG, PP and CTF. In these foods, the “growth rate” did not differ significantly (*p* ≥ 0.05). On the other hand, the “growth rate” values for IGS and CCM were significantly higher (*p* < 0.05). At the storage temperature of 40 °C, the “growth rate” values were significantly higher (*p* < 0.05) in comparison with the lower storage temperatures for all monitored foods (IGS, SG, CCM, PP and CTF).

In Appendix A in the Appendix A, the contents of 17 amino acids during the two-year storage at temperatures of −18 °C, 5 °C, 25 °C and 40 °C are presented. The performed statistical evaluation showed that, practically, there are three groups of amino acids divided according to temperature sensitivity and shelf life. Valine, isoleucine, leucine, phenylalanine, asparagic acid, glutamic acid, proline, glycine, alanine, histidine and arginine belong to the most stable amino acids, and losses usually range up to 5% rel. calculated on the amino acid contents in Month “0” (beginning of the storage experiment). Further, a group of less stable amino acids—lysine, methionine and cysteine—was identified where losses during the storage usually reached below 10% rel., and thus were generally statistically significant (*p* < 0.05). However, Appendix A in the Appendix A showed a third group of so-called unstable amino acids—threonine, serine and tyrosine—with losses reaching even 15% rel. (*p* < 0.05).

### 3.3. Results of Hardness

The nature of CCM and PP samples allowed penetrometric hardness testing; results are shown in Table 4. When considering the temperatures of 5 °C, 25 °C and 40 °C, the hardness values decreased with the increasing storage temperature (*p* < 0.05) and the extending shelf life (*p* < 0.05) for both monitored foods (CCM and PP). The development of hardness measured as the “decrease rate” was significantly higher (absolute values compared) at the storage temperature of −18 °C, which indicates more intense structural changes in CCM and PP during freezer storage.

## 4. Discussion

The microbiology analysis results show that the thermal sterilisation of hermetically sealed containers provides practically sterile products in which no especially spore-forming bacteria that could germinate and spoil the contents of the product during storage are detected. In contrast, as proved by IGS and the decrease in water activity, dehydration is not a sufficient preservation measure that would provide for quality and safe food during two-year-long storage at temperatures of −18 °C to 40 °C. This corresponds with the literature by Stevenson et al. [32], which states that the decrease in water activity only slows the development of spore-forming bacteria.

The analyses of dry matter, crude protein and fat content further revealed that not only metal containers and aluminium containers with polymer internal coatings but also multi-layer packets provide good barrier properties, especially in the prevention of water evaporation for the whole storage period of 24 months at temperatures of −18 °C to 40 °C. In addition to the practical impact of this finding, this phenomenon had another result, namely the conservation of similar dry matter, crude protein and fat content, which enables objective comparison of products during the period of time [2].

The amino acid analysis showed that there are three groups of amino acids divided according to their sensitivity to storage conditions (temperature and storage time). Phenylalanine, asparagic acid, glutamic acid, proline, glycine, alanine, histidine and arginine were identified as very stable amino acids, followed by valine, isoleucine and leucine. The last three amino acids tend to be problematic with respect to their release from more complex (usually vegetable) matrices. This corresponds with other studies, such as Fountoulakis and Lahm [33] and Sarwar et al. [34]. On the contrary, amino acids with the hydroxy group (threonine, serine and tyrosine) contain sulphur in the molecule (methionine and cysteine) and, finally, lysine, whose amino group in ε position is considered reactive, were determined as less stable or unstable. Refs. [33,34,35,36,37] also correspond to these conclusions. With these amino acids, losses of 5 to 15% rel., depending on the applied temperature and the length of storage, were determined.

The indirect proof of amino acid losses and, more generally, the reaction course of nitrogenous substances is also represented by the increase in the content of ammonia during the two-year-long storage. The rate of change grew with the increasing storage temperature. The same conclusions were also reached in the works [2,16,17,35,36]. Based on the growth of the content of ammonia, the course of Maillard reactions and the Strecker degradation of amino acids can be assumed, even at temperatures of <40 °C [13,16,38].

The reactions of nitrogenous substances are not completely stopped by freezer temperatures (applied temperature of −18 °C), which corresponds with, for example, the works of Holman et al. [38] and Pinheiro et al. [13]. It can also be assumed that some intermediate products and final products of Maillard reactions and the Strecker degradation of amino acids (especially carbonylic compounds) also affect the organoleptic properties of food, as was also proved in this work (appearance, consistency, flavour, off-flavour).

During the two-year storage at four different temperatures (−18 °C, 5 °C, 25 °C and 40 °C), oxidative reactions of the lipid fraction were also noted. The contents of the secondary products of lipid oxidation were determined using the TBARS-value, which is an approach recommended in the works of Kristensen et al. [10] and Kristensen and Skibsted [11]. TBARS-values grew along with the extending length of storage, and the rate of growth positively correlated with the storage temperature. The oxidative reactions of lipids probably also caused the worsening of organoleptic flavour and off-flavour properties of CCM stored at 25 °C and 40 °C. These temperatures significantly support oxidative reactions of lipid fractions. Our findings are consistent with, for example, the work of Babji et al. [39], Gokalp et al. [40] and Gomes et al. [41].

The two-year storage substantially affected the consistency of the five products monitored (IGS, SG, CCM, PP and CTF). In the case of CCM and PP, the gradual consistency deterioration can be attributed, among other factors, to the hardness decrease, which could be caused by reactions of nitrogenous substances and possible limited protein cleavage (due to chemical reactions or residual enzyme activity) [16,42,43].

If we summarise the overall results of the performed analyses and subject them to a critical view with respect to the quality and health safety of studied foods (IGS, SG, CCM, PP and CTF), we can conclude that at certain storage temperatures, a critical period during which foods preserve their typical properties exists. Exceeding this period results in a substantial threat to the quality and the health safety of these products. Combinations of the storage temperature and the critical storage period are stated in Table 5.

## 5. Conclusions

Microbiological, chemical, physical and organoleptic changes in long-life canned food and dehydrated food stored at four different temperatures (−18 °C, 5 °C, 25 °C, 40 °C) for the period of 24 months were assessed. The monitored samples of canned food were microbiologically stable at all storage temperatures. In contrast, in the case of IGS stored at the temperature of 40 °C, an intense growth of indicator microorganism groups (*p* < 0.05) was observed to such an extent that the food could not be considered harmless after six months. In all foods, the contents of dry matter, fat and proteins did not change during storage, regardless of the storage temperature (*p* ≥ 0.05). The content of ammonia and the TBARS-value gradually increased in all foods; the more intensively, the higher the storage temperature (*p* < 0.05). The losses of individual amino acids during storage ranged from 5% rel. calculated on the amino acid contents in Month “0” up to 15% rel. (*p* < 0.05). The hardness value has decreased together with the extending storage period, with the highest values for the samples stored at the temperature of −18 °C (*p* < 0.05).

Finally, an evaluation was performed in terms of the quality and health safety of the studied foods (IGS, SG, CCM, PP and CTF). It was ascertained that at certain storage temperatures, a critical period during which foods preserve their typical properties exists. Exceeding the period results in a substantial threat to the quality and health safety of these products.

## Figures and Tables

**Table 1 foods-11-02004-t001:** Result of the microbiological analysis of instant goulash soup during a 24-month period at four different temperatures (−18 °C; 5 °C; 25 °C and 40 °C). The results are expressed as means ± standard deviation (*n* = 27) *.

Storage Time	Storage Temperature	Total Bacterial Count	Aerobic Spore-Forming Bacteria	Anaerobic Spore-Forming Bacteria
(Months)	(°C)	(log CFU/g)	(log CFU/g)	(log CFU/g)
0	-	3.84 ± 0.34 A	2.06 ± 0.19 A	1.65 ± 0.15 A
1	40	5.88 ± 0.55 B	3.17 ± 0.29 B	2.11 ± 0.19 B
3	–18	3.80 ± 0.34 ^a^A	2.03 ± 0.18 ^a^A	1.67 ± 0.15 ^a^A
	5	4.83 ± 0.44 ^b^B	2.50 ± 0.22 ^b^B	1.95 ± 0.17 ^a^A,B
	25	5.07 ± 0.47 ^b^B	2.95 ± 0.27 ^b^B	2.36 ± 0.21 ^b^B
	40	7.55 ± 0.69 ^c^C	3.43 ± 0.30 ^c^B	2.64 ± 0.24 ^b^C
6	5	5.33 ± 0.49 ^a^B	2.66 ± 0.23 ^a^C	2.12 ± 0.20 ^a^B
	25	6.82 ± 0.61 ^b^C	3.10 ± 0.28 ^b^B	2.55 ± 0.23 ^a^B
	40	11.01 ± 1.26 ^c^D	6.22 ± 0.56 ^c^C	4.59 ± 0.42 ^b^D
12	–18	4.56 ± 0.41 ^a^B	2.41 ± 0.21 ^a^B	1.96 ± 0.17 ^a^B
	5	5.67 ± 0.52 ^b^B	3.31 ± 0.29 ^b^D	2.50 ± 0.23 ^b^B
	25	9.69 ± 0.89 ^c^D	4.40 ± 0.40 ^c^C	3.28 ± 0.31 ^c^C
18	5	6.59 ± 0.59 ^a^C	3.01 ± 0.27 ^a^D	2.35 ± 0.21 ^a^B
	25	10.51 ± 1.10 ^b^D	6.45 ± 0.59 ^b^D	4.68 ± 0.43 ^b^D
24	–18	5.73 ± 0.53 ^a^C	2.91 ± 0.26 ^a^C	2.18 ± 0.20 ^a^B
	5	9.22 ± 0.83 ^b^C	4.00 ± 0.34 ^b^E	3.29 ± 0.29 ^b^C

* The means within a column (the difference between the storage temperature at constant storage time) followed by different superscript letters differ (*p* < 0.05). The means within a column (the difference between the storage period at the same temperature) followed by different capital letters differ (*p* < 0.05).

**Table 2 foods-11-02004-t002:** Result of the dry matter content (% *w*/*w*), crude protein content (% *w*/*w*) and fat content (% *w*/*w*) during 24-month period at four different temperatures (−18 °C, 5 °C, 25 °C and 40 °C). Instant goulash soup, Szeged goulash, canned chicken meat, pork pâté and canned tuna fish were tested. The results are expressed as means ± standard deviation (*n* = 27) *.

Sample	Storage Time	Storage Temperature	Dry Matter Content	Crude Protein Content	Fat Content
	(Months)	(°C)	(% *w*/*w*)	(% *w*/*w*)	(% *w*/*w*)
Instant	0	-	96.30 ± 4.53 A	13.26 ± 0.58 A	11.78 ± 0.55 A
goulash	12	–18	95.81 ± 5.55 ^a^A	13.08 ± 0.65 ^a^A	12.26 ± 0.60 ^a^A
soup		5	95.00 ± 4.65 ^a^A	13.01 ± 0.73 ^a^A	12.16 ± 0.61 ^a^A
		25	96.03 ± 4.98 ^a^A	12.84 ± 0.68 ^a^A	11.62 ± 0.65 ^a^A
		40	95.68 ± 5.30 ^a^A	13.03 ± 0.60 ^a^A	12.07 ± 0.69 ^a^A
	24	–18	95.82 ± 4.57 ^a^A	12.94 ± 0.66 ^a^A	12.04 ± 0.59 ^a^A
		5	94.55 ± 4.04 ^a^A	13.27 ± 0.65 ^a^A	11.98 ± 0.59 ^a^A
		25	96.47 ± 5.71 ^a^A	13.12 ± 0.61 ^a^A	11.93 ± 0.62 ^a^A
Szeged	0	-	18.39 ± 1.05 ^a^A	4.47 ± 0.23 ^a^A	9.50 ± 0.53 ^a^A
goulash	12	–18	17.60 ± 0.88 ^a^A	4.54 ± 0.21 ^a^A	9.54 ± 0.41 ^a^A
		5	18.55 ± 1.07 ^a^A	4.60 ± 0.22 ^a^A	9.28 ± 0.44 ^a^A
		25	18.16 ± 0.82 ^a^A	4.36 ± 0.23 ^a^A	9.52 ± 0.50 ^a^A
		40	17.97 ± 0.79 ^a^A	4.39 ± 0.21 ^a^A	10.04 ± 0.39 ^a^A
	24	–18	18.03 ± 0.82 ^a^A	4.57 ± 0.26 ^a^A	9.39 ± 0.48 ^a^A
		5	18.44 ± 0.85 ^a^A	4.60 ± 0.24 ^a^A	9.25 ± 0.43 ^a^A
		25	18.05 ± 0.98 ^a^A	4.47 ± 0.18 ^a^A	9.06 ± 0.49 ^a^A
		40	18.00 ± 0.95 ^a^A	4.48 ± 0.21 ^a^A	9.61 ± 0.35 ^a^A
Canned	0	-	28.24 ± 1.35 ^a^A	17.02 ± 0.90 ^a^A	4.86 ± 0.26 ^a^A
chicken	12	–18	28.83 ± 1.14 ^a^A	17.55 ± 0.86 ^a^A	4.83 ± 0.18 ^a^A
meat		5	28.48 ± 1.38 ^a^A	16.64 ± 0.91 ^a^A	4.91 ± 0.25 ^a^A
		25	28.97 ± 1.40 ^a^A	16.62 ± 0.85 ^a^A	5.19 ± 0.27 ^a^A
		40	27.56 ± 1.36 ^a^A	16.70 ± 0.78 ^a^A	5.03 ± 0.26 ^a^A
	24	–18	28.41 ± 1.41 ^a^A	16.83 ± 0.84 ^a^A	4.90 ± 0.24 ^a^A
		5	28.09 ± 1.23 ^a^A	17.28 ± 0.85 ^a^A	4.95 ± 0.26 ^a^A
		25	29.68 ± 1.42 ^a^A	17.14 ± 0.86 ^a^A	5.15 ± 0.26 ^a^A
		40	28.60 ± 1.54 ^a^A	17.15 ± 0.98 ^a^A	5.24 ± 0.29 ^a^A
Pork	0	-	46.02 ± 2.24 ^a^A	28.63 ± 1.38 ^a^A	8.26 ± 0.53 ^a^A
paté	12	–18	45.10 ± 2.41 ^a^A	28.13 ± 1.42 ^a^A	8.81 ± 0.39 ^a^A
		5	46.24 ± 2.15 ^a^A	28.21 ± 1.32 ^a^A	9.21 ± 0.54 ^a^A
		25	46.20 ± 1.89 ^a^A	29.24 ± 1.50 ^a^A	9.07 ± 0.44 ^a^A
		40	45.25 ± 2.34 ^a^A	28.65 ± 1.34 ^a^A	9.03 ± 0.51 ^a^A
	24	–18	45.92 ± 2.31 ^a^A	28.61 ± 1.43 ^a^A	9.12 ± 0.42 ^a^A
		5	45.38 ± 2.38 ^a^A	29.52 ± 1.42 ^a^A	8.83 ± 0.44 ^a^A
		25	45.11 ± 1.94 ^a^A	29.13 ± 1.48 ^a^A	9.16 ± 0.53 ^a^A
		40	46.54 ± 2.23 ^a^A	29.45 ± 1.64 ^a^A	9.13 ± 0.35 ^a^A
Canned	0	-	20.22 ± 1.01 ^a^A	17.90 ± 0.91 ^a^A	7.21 ± 0.34 ^a^A
tuna	12	–18	20.92 ± 1.01 ^a^A	17.56 ± 0.67 ^a^A	7.03 ± 0.44 ^a^A
fish		5	20.45 ± 1.04 ^a^A	18.27 ± 0.98 ^a^A	7.07 ± 0.38 ^a^A
		25	21.22 ± 1.11 ^a^A	18.51 ± 0.83 ^a^A	6.94 ± 0.33 ^a^A
		40	21.65 ± 1.46 ^a^A	18.26 ± 0.81 ^a^A	6.98 ± 0.36 ^a^A
	24	–18	21.03 ± 0.95 ^a^A	18.47 ± 0.89 ^a^A	6.94 ± 0.44 ^a^A
		5	20.54 ± 1.06 ^a^A	18.00 ± 0.81 ^a^A	6.96 ± 0.33 ^a^A
		25	21.19 ± 1.11 ^a^A	18.62 ± 0.95 ^a^A	6.88 ± 0.32 ^a^A
		40	21.84 ± 0.65 ^a^A	17.56 ± 0.94 ^a^A	7.18 ± 0.34 ^a^A

* The means within a column (the difference between the storage temperature at constant storage time) followed by different superscript letters differ (*p* < 0.05). The means within a column (the difference between the storage period at the same temperature) followed by different capital letters differ (*p* < 0.05).

**Table 3 foods-11-02004-t003:** Result of the ammonia content (mg·kg^−1^) (% *w*/*w*) and TBARS-value (A_538_·mg^−1^) during 24-month period at four different temperatures (−18 °C, 5 °C, 25 °C and 40 °C). Instant goulash soup, Szeged goulash, canned chicken meat, pork pâté and canned tuna fish were tested. The results are expressed as means ± standard deviation (*n* = 27) *.

Sample	Storage Time	Storage Temperature	Ammonia Content	TBARS-Value
	(Months)	(°C)	(mg/kg)	(A_538_/mg)
Instant	0	-	39.9 ± 0.8 A	136.2 ± 3.1 A
goulash	1	40	95.2 ± 2.7 B	157.1 ± 3.2 B
soup	3	–18	101.4 ± 2.9 ^a^B	143.5 ± 3.1 ^a^B
		5	128.2 ± 3.0 ^b^B	144.3 ± 3.2 ^a^B
		25	152.3 ± 3.2 ^c^B	154.8 ± 3.3 ^b^B
		40	161.8 ± 3.6 ^d^C	306.7 ± 5.6 ^c^C
	6	5	147.1 ± 3.6 ^a^C	161.3 ± 3.6 ^a^C
		25	168.2 ± 3.7 ^b^C	169.1 ± 3.7 ^a^C
		40	194.2 ± 3.9 ^c^D	351.9 ± 4.9 ^c^D
	9	5	155.3 ± 3.3 ^a^D	180.4 ± 3.9 ^a^D
		25	176.1 ± 3.8 ^b^D	191.1 ± 4.0 ^b^D
		40	249.5 ± 5,7 ^c^E	401.7 ± 6,1 ^c^E
	12	–18	130.9 ± 3.1 ^a^C	150.4 ± 3.7 ^a^C
		5	169.4 ± 3.7 ^b^E	189.0 ± 4.2 ^b^E
		25	188.2 ± 3.9 ^c^E	203.8 ± 4.8 ^c^E
		40	NE **	NE
	15	5	175.4 ± 2.6 ^a^F	199.3 ± 3.3 ^a^F
		25	206.3 ± 3.4 ^b^F	206.7 ± 2.1 ^b^F
		40	NE	NE
	18	5	182.9 ± 3.9 ^a^G	197.7 ± 4.6 ^a^F
		25	215.8 ± 3.5 ^b^G	203.6 ± 3.0 ^b^F
		40	NE	NE
	21	5	190.9 ± 3.7 H	192.5 ± 2.7 F
		25	NE	NE
		40	NE	NE
	24	–18	190.4 ± 3.1 ^a^D	177.4 ± 3.3 ^a^D
		5	205.5 ± 3.4 ^b^I	185.4 ± 4.1 ^b^E
		25	NE	NE
		40	NE	NE
Szeged	0	-	45.7 ± 1.0 A	167.0 ± 3.3 A
goulash	1	40	55.9 ± 1.1 B	241.4 ± 4.7 B
	3	–18	48.2 ± 1.0 ^a^B	208.8 ± 4.1 ^a^B
		5	50.8 ± 1.0 ^b^B	218.4 ± 4.2 ^b^B
		25	53.3 ± 1.1 ^c^B	236.1 ± 4.6 ^c^B
		40	60.9 ± 1.3 ^d^C	299.7 ± 5.0 ^d^C
	6	5	54.1 ± 1.2 ^a^C	278.1 ± 5.2 ^a^C
		25	56.5 ± 1.2 ^a^C	302.7 ± 5.6 ^b^C
		40	66.3 ± 1.4 ^b^D	357.1 ± 6.1 ^c^D
	9	5	57.2 ± 1.2 ^a^D	305.9 ± 6.0 ^a^D
		25	59.7 ± 1.3 ^a^D	349.2 ± 6.4 ^b^D
		40	82.5 ± 3.2 ^b^D	409.5± 8.3 ^c^D
	12	–18	52.4 ± 1.1 ^a^C	261.4 ± 5.1 ^a^C
		5	60.1 ± 1.3 ^b^E	342.0 ± 6.4 ^b^E
		25	61.7 ± 1.3 ^b^D	391.5 ± 6.7 ^c^E
		40	98.6 ± 1.4 ^c^E	438.7 ± 7.2 ^d^E
	15	5	63.3 ± 1.4 ^a^F	330.1 ± 3.9 ^a^F
		25	68.2 ± 0.8 ^b^E	374.2 ± 3.1 ^b^F
		40	117.0 ± 7.3 ^c^F	458.9 ± 4.8 ^c^F
	18	5	58.4 ± 0.4 ^a^D,E	301.0 ± 1.2 ^a^D
		25	70.8 ± 1.2 ^b^F	359.2 ± 4.3 ^b^G
		40	143.1 ± 8.1 ^c^G	491.1 ± 5.9 ^c^F
	21	5	76.2 ± 1.4 ^a^F	301.8 ± 2.9 ^a^D
		25	81.2 ± 0.8 ^b^G	345.8 ± 4.8 ^b^D
		40	177.2 ± 9.8 ^c^H	539.5 ± 6.3 ^c^G
	24	–18	63.0 ± 1.0 ^a^D	257.7 ± 4.2 ^a^D
		5	63.5 ± 2,3 ^a^G	327.5 ± 4,5 ^b^E
		25	98.4 ± 2.1 ^b^H	391.1 ± 5.3 ^c^E
		40	201.9 ± 9.7 ^c^I	609.4 ± 6.6 ^d^I
Canned	0	-	71.1 ± 1.5 A	6.7 ± 0.1 A
chicken	1	40	93.9 ± 1.8 B	13.2 ± 0.3 B
meat	3	–18	88.9 ± 1.5 ^a^B	11.9 ± 0.2 ^a^B
		5	91.4 ± 1.7 ^b^B	12.3 ± 0.2 ^b^B
		25	93.2 ± 1.7 ^b^B	15.5 ± 0.2 ^c^B
		40	106.6 ± 2.0 ^c^C	29.8 ± 0.6 ^d^C
	6	5	99.6 ± 1.9 ^a^C	24.7 ± 0.7 ^a^C
		25	108.7 ± 2.0 ^b^C	30.8 ± 0.7 ^b^C
		40	115.9 ± 2.2 ^c^D	58.6 ± 0.9 ^c^D
	9	5	110.7 ± 2.1 ^a^D	35.7 ± 0.8 ^a^D
		25	113.8 ± 2.1 ^a^D	50.1 ± 0.9 ^b^D
		40	149.5 ± 3.8 ^b^D	69.2 ± 2.7 ^c^D
	12	–18	95.4 ± 1.7 ^a^C	31.2 ± 0.6 ^a^C
		5	115.9 ± 2.1 ^b^E	44.4 ± 0.7 ^b^E
		25	121.9 ± 2.3 ^c^E	62.8 ± 0.9 ^c^E
		40	NE **	NE
	15	5	121.7 ± 1.9 ^a^F	46.3 ± 0.1 ^a^F
		25	126.6 ± 1.7 ^b^F	64.7 ± 0.3 ^b^F
		40	NE	NE
	18	5	136.8 ± 2.2 ^a^G	41.6 ± 2.1 ^a^G
		25	157.4 ± 1.4 ^b^G	57.1 ± 1.1 ^b^G
		40	NE	NE
	21	5	152.3 ±1.4 ^a^H	38.5 ± 0.8 ^a^G
		25	198.0 ± 2.1 ^b^H	55.6 ± 1.3 ^b^G
		40	NE	NE
	24	–18	107.3 ± 1.8 ^a^D	40.3 ± 1.5 ^a^D
		5	176.4 ± 2.7 ^b^I	49.3 ± 1.8 ^b^I
		25	NE	NE
		40	NE	NE
Pork	0	-	30.5 ± 0.6 A	28.2 ± 0.6 A
paté	1	40	53.4 ± 1.1 B	39.7 ± 0.8 B
	3	–18	40.6 ± 0.7 ^a^B	32.5 ± 0.7 ^a^B
		5	43.2 ± 0.5 ^b^B	37.6 ± 0.7 ^b^B
		25	45.7 ± 0.7 ^b^B	41.7 ± 0.7 ^c^B
		40	57.1 ± 1.0 ^c^C	61.9 ± 1.5 ^d^C
	6	5	47.9 ± 0.8 ^a^C	44.6 ± 0.8 ^a^C
		25	50.6 ± 0.8 ^b^C	49.4 ± 0.8 ^b^C
		40	62.4 ± 1.1 ^c^D	73.1 ± 1.6 ^c^D
	9	5	51.0 ± 0.9 ^a^D	51.2 ± 0.9 ^a^D
		25	55.4 ± 1.1 ^b^D	58.4 ± 1.1 ^b^D
		40	76.1 ± 2.4 ^c^E	89.0 ± 2.5 ^c^E
	12	–18	49.7 ± 0.9 ^a^C	41.6 ± 0.7 ^a^C
		5	57.0 ± 1.3 ^b^E	59.0 ± 1.4 ^b^E
		25	60.3 ± 1.3 ^c^E	67.7 ± 1.7 ^c^E
		40	91.6 ± 2.7 ^d^F	102.7 ± 1.6 ^d^F
	15	5	55.8 ± 1.7 ^a^E	62.1 ± 1.5 ^a^F
		25	66.0 ± 1.4 ^b^F	73.2 ± 0.8 ^b^F
		40	109.7 ± 2.2 ^c^G	121.9 ± 2.6 ^c^G
	18	5	60.9 ± 1.8 ^a^F	58.9 ± 1.7 ^a^F
		25	73.6 ± 0.5 ^b^G	70.0 ± 1.4 ^b^F
		40	125.9 ± 3.1 ^c^H	139,9 ± 3.4 ^c^H
	21	5	63.2 ± 1.5 ^a^G	56.4 ± 1.0 ^a^F
		25	79.2 ± 0.9 ^b^H	66.7 ± 1.2 ^b^G
		40	139.1 ± 3.0 ^c^I	147.2 ± 2.8 ^c^I
	24	–18	60.9 ± 1.0 ^a^D	44.5 ± 1.3 ^a^C
		5	67.5 ± 0.4 ^b^H	55.8 ± 1.4 ^b^F
		25	83.2 ± 0.6 ^c^I	67.4 ± 1.7 ^c^G
		40	151.9 ± 2.7 ^d^J	154.8 ± 1.9 ^d^J
Canned	0	-	68.5 ± 1.4 A	30.8 ± 0.6 A
tuna	1	40	76.2 ± 1.5 B	56.2 ± 1.4 B
fish	3	–18	70.1 ± 1.4 ^a^A	31.1 ± 0.6 ^a^A
		5	72.3 ± 1.4 ^a^B	53.3 ± 1.3 ^b^B
		25	79.2 ± 1.5 ^b^B	56.2 ± 1.4 ^b^B
		40	88.6 ± 1.6 ^c^C	77.2 ± 1.6 ^c^C
	6	5	76.3 ± 1.5 ^a^C	62.1 ± 1.4 ^a^C
		25	84.0 ± 1.8 ^b^C	67.8 ± 1.5 ^b^C
		40	97.0 ± 1.9 ^c^D	89.5 ± 1.9 ^c^D
	9	5	80.9 ± 1.6 ^a^D	68.8 ± 1.6 ^a^D
		25	87.3 ± 1.7 ^b^D	72.1 ± 1.6 ^b^D
		40	103.7 ± 1.5 ^c^E	92.6 ± 1.8 ^c^E
	12	–18	74.0 ± 1.6 ^a^B	53.4 ± 1.3 ^a^C
		5	85.3 ± 1.7 ^b^E	75.4 ± 1.7 ^b^E
		25	92.7 ± 1.9 ^c^E	78.3 ± 1.8 ^c^E
		40	109.2 ± 2.4 ^d^F	97.0 ± 1.9 ^d^F
	15	5	83.8 ± 0.9 ^a^E	79.6 ± 1.6 ^a^F
		25	96.5 ± 1.5 ^b^F	83.6 ± 1.3 ^b^F
		40	117.2 ± 1.9 ^c^G	106.3 ± 2.0 ^c^G
	18	5	96.2 ± 1.3 ^a^F	76.9 ± 1.6 ^a^E
		25	102.7 ± 1.4 ^b^G	79.7 ± 1.9 ^b^E
		40	131.8 ± 2.8 ^c^H	120.4 ± 2.3 ^c^H
	21	5	107.1 ± 1.8 ^a^G	73.3 ± 2.0 ^a^E
		25	115.3 ± 1.7 ^b^H	76.7 ± 1.5 ^b^E
		40	139.3 ± 1.6 ^c^I	128.5 ± 2.7 ^c^I
	24	–18	76.9 ± 1.8 ^a^C	60.0 ± 1.1 ^a^D
		5	112.3 ± 1.4 ^b^H	70.9 ± 1.2 ^b^D
		25	126.4 ± 1.5 ^c^I	73.1 ± 1.9 ^c^D
		40	149.8 ± 2.9 ^d^J	137.2 ± 2.7 ^d^J

* The means within a column (the difference between the storage temperature at constant storage time) followed by different superscript letters differ (*p* < 0.05). The means within a column (the difference between the storage period at the same temperature) followed by different capital letters differ (*p* < 0.05). All samples were evaluated separately. ** NE—not evaluated due to the results of sensory analysis and/or microbiological analysis.

**Table 4 foods-11-02004-t004:** Results of hardness (N) of canned chicken meat and pork pâté dependent on various temperature regimens (−18 °C; 5 °C; 25 °C; and 40 °C) and storage time (months). The results are presented as means ± standard deviations (*n* = 27) *.

Storage Time	Storage Temperature	Canned Chicken	Pork Paté
(Months)	(°C)	Meat	
0	-	119.3 ± 6.0 A	62.8 ± 2.7 A
1	40	117.4 ± 5.4 B,C	62.1 ± 3.6 A
3	–18	97.9 ± 5.3 ^a^B	58.3 ± 3.3 ^a^B
	5	117.5 ± 5.2 ^b^A	63.1 ± 2.7 ^b^A
	25	115.9 ± 5.0 ^b,c^B	64.5 ± 3.1 ^b^A
	40	113.4 ± 5.5 ^c^C	58.2 ± 2.9 ^c^B
6	5	115.9 ± 5.4 ^a^B	55.2 ± 3.2 ^a^B
	25	108.6 ± 5.0 ^b^A	53.4 ± 3.2 ^a^B
	40	92.2 ± 4.1 ^c^D	50.9 ± 2.6 ^b^C
9	5	115.0 ± 5.7 ^a^B	48.3 ± 2.7 ^a^C
	25	102.4 ± 5.1 ^b^C	46.6 ± 2.7 ^a^C
	40	81.7 ± 4.3 ^c^E	43.5 ± 2.0 ^b^D
12	–18	84.9 ± 4.7 ^a^C	33.3 ± 2.0 ^a^C
	5	100.3 ± 6.3 ^b^C	43.2 ± 1.8 ^b^D
	25	91.7 ± 4.8 ^c^D	40.1 ± 1.4 ^b,c^E
	40	64.4 ± 3.0 ^d^F	35.7 ± 1.9 ^a,c^E
15	5	98.2 ± 5.6 ^a^C	39.6 ± 2.1 ^a^E
	25	87.9 ± 5.0 ^b^E	36.6 ± 1.6 ^b^F
	40	NE **	31.9 ± 1.6 ^c^F
18	5	95.8 ± 4.3 ^a^C,D	36.7 ± 1.8 ^a^E,F
	25	80.0 ± 3.6 ^b^F	33.1 ± 1.6 ^b^G
	40	NE	26.0 ± 1.0 ^c^G
21	5	89.7 ± 5.0 ^a^D	34.4 ± 1.7 ^a^F
	25	77.9 ± 3.5 ^b^G	29.9 ± 1.3 ^b^H
	40	NE	19.4 ± 0.9 ^c^H
24	–18	62.7 ± 3.7 ^a^D	25.7 ± 1.2 ^a^D
	5	82.8 ± 3.9 ^b^E	30.7 ± 1.5 ^b^G
	25	NE	28.4 ± 1.2 ^b^I
	40	NE	15.6 ± 0.7 ^c^I

* The means within a column (the difference between the storage temperature at constant storage time) followed by different superscript letters differ (*p* < 0.05). The means within a column (the difference between the storage period at the same temperature) followed by different capital letters differ (*p* < 0.05). ** NE—not evaluated due to the results of sensory analysis.

**Table 5 foods-11-02004-t005:** Combination of certain storage temperatures (−18 °C, 5 °C, 25 °C and 40 °C) and critical time (months) for preserving food quality and safety of five foods tested (instant goulash soup, Szeged goulash, canned chicken meat, pork pâté and canned tuna fish).

Sample	Temperature			
	−18 °C	5 °C	25 °C	40 °C
Instant goulash soup	<24 months	≤24 months	≤15 months	<6 months
Szeged goulash	≤24 months	≤18 months	≤18 months	≤18 months
Canned chicken meat	≤24 months	≤18 months	≤18 months	≤3 months
Pork paté	≤24 months	≤24 months	≤24 months	≤18 months
Canned tuna fish	≤24 months	≤24 months	≤24 months	≤9 months

## Data Availability

The data presented in this study are available on request from the corresponding author.

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
