# Peer review of "Changes in the Quality Attributes of Selected Long-Life Food at Four Different Temperatures over Prolonged Storage"

_foods, 2022, doi:10.3390/foods11142004_

Round 1

Reviewer 1 Report

The paper is novel, interesting and valuable. I have only a few comments.

Line 117. "x" is missing twice in the equation.

Lines 141 and 148. When were these analyses performed? Please specify.

Line 259. How was the rate of increase calculated? Please specify.

Lines 262-266. It seems confusing. Please rewrite these sentences and please make them shorter.

Line 391. Please add S to Table 2 – Table S2.

Reviewer 2 Report

In the manuscript entitled “Changes in the quality attributes of selected long-life food at four different temperatures over prolonged storage”, the authors report the development of selected indicators affecting changes in food quality and safety of selected long-life food during a prolonged storage experiment at four different temperatures.

The manuscript is well written, results are well presented and supported with adequate experimental data.

However, the manuscript could be improved by adding some references (values or cut-off limits) for the particular analyzed parameter if they exist.

In lines 120-121 the terms “too-high microbiological activity” and “failed sensory assessment” should be explained in more detail.

The chapter “2.6 Determination of amino acids” should be more precise. At least some experimental conditions (column, instrument, mobile phase) and use of standards (purchase, purity,…) should be added.

In the manuscript entitled “Changes in the quality attributes of selected long-life food at four different temperatures over prolonged storage”, the authors report the development of selected indicators affecting changes in food quality and safety of selected long-life food during a prolonged storage experiment at four different temperatures.

The manuscript is well written, results are well presented and supported with adequate experimental data.

However, the manuscript could be improved by adding some references (values or cut-off limits) for the particular analyzed parameter if they exist.

In lines, 120-121 the terms “too-high microbiological activity” and “failed sensory assessment” should be explained in more detail.

The chapter “2.6 Determination of amino acids” should be more explicit. At least some experimental conditions (column, instrument, mobile phase) and use of standards (purchase, purity,…) should be added.
